# Identification of families in need of support: Correlates of adverse childhood experiences in the right@home sustained nurse home visiting program

Lynn Kemp[1]*, Tracey Bruce[1], Emma L. Elcombe[1], Fiona Byrne[1], Sheryl A. Scharkie[1], Susan M. Perlen[2], Sharon R. Goldfeld[2,3,4]

1 School of Nursing and Midwifery, Western Sydney University, Ingham Institute for Applied Medical Research, Liverpool, New South Wales, Australia, 2 Population Health, Murdoch Children's Research Institute, Parkville, Victoria, Australia, 3 Centre for Community Child Health, The Royal Children's Hospital, Parkville, Victoria, Australia, 4 Department of Paediatrics, University of Melbourne, Parkville, Victoria, Australia

☯ These authors contributed equally to this work.
* Lynn.Kemp@westernsydney.edu.au

**Data Availability Statement:** Data cannot be shared publicly because they contain sensitive participant information which are restricted to use for research purposes only. This restriction is in

## Abstract

### Background

Little is known about the efficacy of pregnancy screening tools using non-sensitive sociodemographic questions to identify the possible presence of as yet undiagnosed disease in individuals and later adverse childhood events disclosure.

### Objectives

The study aims were to: 1) record the prevalence of risk disclosed by families during receipt of a sustained nurse home visiting program; and 2) explore patterns of relationships between the disclosed risks for their child having adverse experiences and the antenatal screening tool, which used non-sensitive demographic questions.

### Design

Retrospective, observational study.

### Participants and methods

Data about the participants in the intervention arm of the Australian right@home trial, which is scaffolded on the Maternal Early Childhood Sustained Home-visiting model, collected between 2013 and 2017 were used. Screening data from the 10-item antenatal survey of non-sensitive demographic risk factors and disclosed risks recorded by the nurse in audited case files during the subsequent 2 year intervention were examined (n = 348). Prevalence of disclosed risks for their child having adverse experiences were analysed in 2019 using multiple response frequencies. Phi correlations were conducted to test associations between screening factors and disclosed risks.

accordance with the Participant Information and Consent Form and approved study protocol, governed by the Royal Children's Hospital Human Research Ethics Committee (HREC 32296). We invite researchers to request access to the data from the Melbourne Children's Campus LifeCourse institutional data access platform (https://lifecourse.melbournechildrens.com/data-access/) or the governing Royal Children's Hospital HREC (https://www.rch.org.au/ethics/).

**Funding:** This work is supported by the Victorian Department of Education and Early Childhood Development, the Tasmanian Department of Health, the Ian Potter Foundation, Sabemo Trust, Sidney Myer Fund, the Vincent Fairfax Family Foundation, and the National Health and Medical Research Council (NHMRC, 1079418). The funders had no role in study design, data collection and analysis, decision to publish, or preparation of the manuscript.

**Competing interests:** The authors have declared that no competing interests exist.

## Results

Among the 348 intervention participants whose files were audited, 300 were noted by nurses to have disclosed risks during the intervention, with an average of four disclosures. The most prevalent maternal disclosures were depression or anxiety (57.8%). Mental health issues were the most prevalent partner and family disclosures. Screening tool questions on maternal smoking in pregnancy, not living with another adult, poverty and self-reporting anxious mood were significantly associated with a number of disclosed risks for their child having adverse experiences.

## Conclusions

These findings suggest that a non-sensitive sociodemographic screening tool may help to identify families at higher risk for adverse childhood experiences for whom support from a sustained nurse home visiting program may be beneficial.

## Introduction

Adverse childhood experiences (ACEs), such as exposure to child abuse, neglect and household dysfunction, are detrimental for child and adult health. Felitti [1] in a large population study noted linear relationships between adults' recollection of the number of ACEs they were exposed to, and adult health issues and mortality, with lifelong costs and consequences [2–4]. Recent research has shown that more proximal recall of ACEs has been associated with the prevalence of physical, mental and developmental health conditions in children and young people aged 0–17 years [2, 5].

Children's exposure to ACEs is now seen as a priority public health issue and strategies to reduce exposure are needed. One suggested strategy is screening parents of very young children for ACEs [6]. Policies of screening for psychosocial risks have been implemented in Australia [7] and other countries [8]. There is evidence, however, that women find questioning about sensitive issues such as their own ACEs or current household dysfunction, particularly family violence and drug and alcohol issues, to be distressing [9], and for those with present risk, daunting and intrusive [9]. Some do not disclose risks as they do not think it would lead to help [8], or provide socially acceptable, rather than truthful, responses, and some do not disclose due to fear of stigma or negative consequences such as loss of child custody [10, 11]. For example, Johnson [12] suggested low rates of disclosure of alcohol and substance use may be the result of public awareness of the risk these pose in pregnancy and breastfeeding; and a qualitative study notes that 'at least 20 per cent of those who experience IPV [intimate partner violence] tell no one else about it' [13].

Consequently, WHO, whilst supportive of universal screening, does not advocate screening in all women. Rather WHO recommends screening in health care visits when factors known to be associated with ACEs are present, such as depression, self-harm or presence of an intrusive partner [14, 15]. However, this approach would require health care providers to have the opportunity to observe such factors. Surveillance within public health provision is complex with debates over practice and ethics, and the power relationships between clients and clinicians [16, 17]. Clinicians also find universal screening to be unfeasible in postpartum visits [18], and are uncomfortable with routine and standardized questioning about issues such as domestic

violence and abuse; preferring to gather information in diverse ways such as "cloaking' the assessment in the baby check and downplaying the assessment as 'doing some paperwork" [9].

Privacy, a safe and confidential environment and a supportive relationship between the clinician and the client are needed for both inquiry and disclosure of ACEs [16, 18], with the latter necessarily absent in universal screening. There is evidence that when screening occurs in an established relationship parents feel more comfortable to provide honest information in the context of a strong working relationship [19]. For example, a study investigating the feasibility of implementing ACE screening of parents found that within the Early Head Start home visiting program group, prevalence of disclosed risk was considerably higher when screening occurred weeks or months into the program, compared to screening at first visit or having minimal contact [12].

A screening process is therefore needed that does not require disclosure of sensitive issues, but can 'identify the possible presence of an as-yet-undiagnosed disease in individuals without signs or symptoms. This can include individuals with presymptomatic or unrecognized symptomatic disease' [14]. Little is known, however, of non-sensitive sociodemographic factors associated with ACE risks subsequently disclosed. A recent US study has suggested that birth certificate data such as acknowledgement of paternity and race may be useful to indicate risks for child maltreatment [20]. Prevention and early intervention programs such as sustained home visiting that commence in pregnancy have relied on bluntly targeting services to easily identifiable population categories, typically young, poor, first-time mothers. Evidence is emerging however, that the population prevalence of risk is not solely (or even largely) concentrated in these (or any) group [21, 22]. The challenge is therefore, ascertaining how prevention and early intervention services can screen and implement programs for families of children at risk of ACEs in ways that do not require them to disclose those risks in order to be considered eligible for the program.

In the context of the Australian right@home randomized controlled trial [23], a screening tool was developed to identify pregnant women who could potentially benefit from additional support to promote positive home environments for their children's health and development and prevent or mitigate the children's exposure to ACEs through receipt of the sustained nurse home visiting program. To facilitate trial recruitment, the screening tool needed to be deliverable in a non-private environment, the antenatal clinic waiting room, and was thus limited to asking non-sensitive questions. Piloting of the tool prior to commencement of the trial demonstrated that it was both feasible and acceptable [24], and was subsequently used as a tool for identifying eligible families for participation in the trial. The objective of this study is to explore risks that were disclosed by families throughout their receipt of the home visiting program and the correlation between these disclosed risks and the non-sensitive screening questions.

## Methods

This observational cohort study was a secondary project based within the intervention arm of the right@home trial [23] to describe the prevalence of disclosed ACE risks and the correlation between specific sociodemographic characteristics in pregnancy and disclosed risk in the context of delivery of the right@home program to child age 2 years.

### right@home program

The right@home program is a sustained nurse home visiting intervention structured around the Maternal Early Childhood Sustained Home-visiting (MECSH) model [25–27] incorporating core content of child development parent education program, social support promotion and group activities with added content modules focussed on language development, parent-

child attachment, healthy eating, child sleeping and home safety [23, 25, 28]. The program improved outcomes in parent care, parent responsivity and the home learning environment [28]. Postgraduate trained nurses delivered the program from the antenatal period until the child's second birthday. The nurses were supported by a social care practitioner who provided instrumental and psychosocial support for families and assisted nurses and families to leverage community resources.

## Sample

722 pregnant women were recruited to the right@home program trial between May 2013 and August 2014 [23, 28]. Eligible women were English speaking, resided within the seven study areas in Victoria and Tasmania Australia, and had two or more of the ten screening risk factors (see Table 1). Women of any age and with any number of children were eligible to participate in the right@home trial. 363 women were randomly allocated to the trial intervention arm, and 352 commenced the intervention program [28]. This secondary study used antenatal screening data and disclosed risk information for mothers who were allocated to the intervention arm of the trial and who received at least one nurse home visit, and whose files were audited (n = 348).

## Data collection and management methods

Antenatal screening was conducted in antenatal clinic waiting rooms by the trial research assistants. At two points during the trial (mid-trial March 2016 and end-trial June 2017), nurses delivering the right@home program to families were asked to conduct a retrospective semi-structured case audit to document risks disclosed by participating mothers. The trial research team provided each nurse with a spreadsheet listing their clients and asked them to note any risks that had been disclosed using predefined drop-down menu categories. Open fields were also provided to capture any risks not in the menu. Information provided was reliant on the accuracy and quality of the records provided by the nurses: no independent audit was conducted and the data provided by the nurses was considered to be complete.

Post collection the text in the open fields was coded into categories by registered nurses in the research team. Multiple disclosures could be noted for each family, and included disclosure of risk associated with the participating mother, her partner and broader family. The coded categories (predefined and other) were summarized into the presence or absence of disclosure of the household challenges identified as contributing to ACEs by the UK 70/30 campaign

**Table 1. Screening data from a 10 question survey: right@home intervention recipients (n = 352).**

| Screening risk factors, maternal report in pregnancy | n (%) |
|---|---|
| Young pregnancy (age < 23 years) | 90 (25.6%) |
| Not living with another adult | 61 (17.3%) |
| No support in pregnancy (financial, emotional or practical) | 31 (8.8%) |
| Poorer health (global health self-reported fair or poor [29]) | 251 (71.3%) |
| Maternal smoking in pregnancy | 115 (32.7%) |
| Long-term illness limiting daily living | 72 (20.5%) |
| Anxious mood (very stressed, anxious, unhappy or difficulty coping in last 2 weeks, and moderately or a lot bothered by the feelings [30]) | 102 (29.0%) |
| School < Year 12: non completion of secondary education in Australia | 193 (54.8%) |
| Poverty (no income in household other than benefits and/or having a means-tested Health Care card) | 123 (34.9%) |
| Never worked | 63 (17.9%) |

[31]: domestic violence, substance abuse, mental illness, parental separation/divorce, and incarcerated parent. Engagement with the child protection system was coded as a proxy for child abuse and neglect.

## Analysis

Prevalence of disclosed ACE risks was analysed in 2019 using multiple response frequencies using SPSS v24.0.0.1. Phi correlation coefficients and associated significance levels were to measure the association between the presence/absence of screening factors and presence/absence of disclosed ACE risks.

## Trial registration

Trial registration number: ISRCTN89962120 (21/Aug/2013), retrospectively registered.

## Compliance with ethical standards

### Ethics approval

The right@home trial was approved by the Human Research Ethics Committees in Australia of: The Royal Children's Hospital, Victoria (HREC 32296); Peninsula Health, Victoria (HREC/13/PH/14); Ballarat Health Services, Victoria (HREC/13/BHSSJOG/9); Southern Health, Victoria (HREC 13084X); Northern Health, Victoria (HREC P03/13); and The University of Tasmania (HREC H0013113). All procedures performed in studies involving human participants were in accordance with the ethical standards of the institutional and/or national research committee and with the 1964 Helsinki declaration and its later amendments or comparable ethical standards. Written consent was obtained.

## Results

No information was received from the nurses for four study participants, who ceased receipt of the intervention early in the program. Most study participants completed the intervention to their child's second birthday (n = 304, 86.4%). The 348 participants in this study were noted by the nurses to have disclosed 969 maternal, 169 partner-related and 167 family related risks: a total of 1305 disclosures over the intervention duration. The mean number of disclosures per participant was four (range 0–16) with 48 participants (13.8%) noted by the nurses as disclosing no risks. The prevalence of risk disclosures categorized as ACEs is presented in Table 2. The most prevalent maternal disclosures were depression (n = 138, 39.4%) and anxiety (n = 120, 34.5%). The most prevalent partner and family disclosure was mental health issues (partner n = 74, 21.3%; family n = 65, 18.5%). Other disclosures not categorized into ACE risks included mother (n = 48), partner (n = 5) and infant (n = 11) physical health concerns, maternal grief (n = 9) and isolation (n = 24). Mothers also disclosed concerns in managing care of disabled or aged family members (n = 17) and financial stress (n = 42). No parental incarceration was recorded.

The correlations between the individual screening questions and mother, partner and family disclosed ACE risk is presented in Fig 1. Maternal smoking in pregnancy, not living with another adult, poverty, and self-reporting anxious mood were significantly associated with the disclosure of a large number of differing ACE risks, although all correlations were weak (less than 0.3). Not completing secondary education was associated with disclosure of partner or family substance misuse, but was also associated with significant non-disclosure of maternal anxiety, depression or mental health conditions.

**Table 2. Prevalence rates of each ACE risk factor disclosed: Nurse audited participants (n = 348).**

| ACE Risk Factors | Disclosure n (%) |
|---|---|
| **Maternal ACE risk factors** | |
| A. Substance misuse | 37 (10.6%) |
| B. Domestic violence | 61 (17.5%) |
| C. Engagement with child protection services for current children | 69 (19.8%) |
| D. Anxiety or Depression | 201 (57.8%) |
| E. Major mental health conditions (bipolar, eating, obsessive-compulsive, personality, post-traumatic stress and psychotic disorders) | 34 (9.8%) |
| F. Any mental health condition (D+E) | 219 (62.9%) |
| G. History of unstable relationships | 62 (17.8%) |
| **Family ACE risk factors** | |
| H. Family/Partner Substance misuse | 65 (18.7%) |
| I. Family/Partner Domestic violence | 15 (4.3%) |
| J. Family/Partner member with Mental health condition (including anxiety or depression or major mental health conditions) | 120 (34.5%) |
| K. Family/Partner with History of unstable relationships | 11 (3.2%) |
| **Combined ACE risk factors** | |
| L. Any Substance Misuse (A+H) | 80 (23.0%) |
| M. Any Domestic Violence (B+I) | 73 (21.0%) |
| N. Any Mental health condition including anxiety or depression (D+E+J) | 240 (69.0%) |
| **O. Any History of unstable relationships (G+K)** | 68 (19.5%) |
| **Number of ACE risk factors disclosed** | |
| **0** | 80 (23.0%) |
| **1** | 110 (31.6%) |
| **2** | 86 (24.7%) |
| **3 or more** | 72 (20.7%) |

## Discussion

The lifelong costs and consequences of early life adversity necessitates early intervention to prevent or mitigate children's exposure to the child abuse, neglect and household dysfunction that are associated with ACEs [3]. Being able to intervene early, however, is premised upon being able to identify those families where risks are present, and there is mounting evidence that reliance on universal screening and first or minimal contact disclosure of sensitive risks is failing to identify those facing adversity and in need of additional support. Knowledge of what non-sensitive characteristics in pregnancy are associated with families' subsequent disclosure of sensitive risks could enable early identification of families in need of additional support and improve the early detection of issues that could lead to children experiencing ACEs. The right@home trial provided an opportunity to explore whether non-sensitive sociodemographic factors in pregnancy, which can be feasibly and acceptably detected through a screening tool administered in public spaces [24], are associated with disclosure of risks that could result in their child/ren having adverse experiences, so that effective early intervention programs can be implemented to improve outcomes.

Recruitment using the non-sensitive screening questionnaire identified an intervention population where ACE risks were highly prevalent. The prevalence in this study sample was considerably higher for all ACEs than the population prevalence reported by parents of children aged 0–17 as recorded by Bright [2], with the rate of disclosed risk more than double in this sample for substance misuse (23.0% cf 10.7%) and domestic violence exposure (21.0% cf

| Disclosures | Risk factors | | | | | | | | | |
|---|---|---|---|---|---|---|---|---|---|---|
| | Young pregnancy | Not living with another adult | No support in pregnancy | Poorer health | Maternal smoking in pregnancy | Long-term illness | Anxious mood | School < Year 12 | Poverty | Never worked |
| **Maternal** | | | | | | | | | | |
| A. Substance Misuse | -0.031 | 0.063 | -0.008 | 0.013 | 0.196** | -0.036 | -0.015 | 0.050 | 0.021 | 0.009 |
| B. Domestic Violence | 0.110* | 0.286** | 0.070 | 0.025 | 0.097 | 0.047 | 0.121* | 0.039 | 0.168** | 0.021 |
| C. Child Protection Services | 0.071 | 0.095 | 0.074 | -0.019 | 0.205** | 0.033 | 0.016 | 0.060 | 0.178** | 0.161** |
| D. Anxiety or Depression | -0.045 | 0.003 | -0.055 | 0.085 | 0.077 | -0.016 | 0.187** | -0.141## | 0.033 | -0.119# |
| E. Other Mental Health condition (major conditions) | 0.007 | 0.155** | 0.000 | -0.005 | 0.059 | 0.120* | -0.039 | 0.026 | 0.083 | -0.027 |
| F. Any Mental Health (D+E) | -0.040 | 0.078 | -0.047 | 0.089 | 0.081 | 0.017 | 0.188** | -0.130# | 0.067 | -0.141## |
| G. Unstable Relationships | 0.054 | 0.143** | 0.041 | 0.095 | -0.036 | -0.087 | 0.116* | 0.030 | 0.099 | -0.021 |
| **Partner or Family** | | | | | | | | | | |
| H. Substance Misuse | 0.090 | 0.111* | -0.019 | -0.054 | 0.199** | -0.114# | -0.030 | 0.123* | 0.127* | -0.069 |
| I. Domestic Violence | 0.005 | -0.022 | -0.016 | -0.115# | 0.093 | 0.032 | -0.073 | 0.050 | -0.037 | -0.062 |
| J. Mental Health condition (any) | 0.046 | -0.060 | -0.033 | 0.072 | -0.003 | 0.066 | 0.029 | -0.022 | -0.024 | -0.039 |
| K. Unstable Relationships | 0.007 | 0.004 | 0.002 | 0.006 | 0.084 | -0.010 | -0.043 | 0.065 | 0.074 | 0.001 |
| **Mother, Partner or Family** | | | | | | | | | | |
| L. Substance Misuse (A+H) | 0.040 | 0.092 | -0.049 | -0.046 | 0.273** | -0.090 | -0.033 | 0.124* | 0.100 | -0.059 |
| M. Domestic Violence (B+I) | 0.086 | 0.247** | 0.064 | -0.016 | 0.137* | 0.053 | 0.090 | 0.070 | 0.139** | -0.001 |
| N. Any Mental Health condition (D+E+J) | -0.019 | 0.055 | -0.046 | 0.066 | 0.112* | -0.001 | 0.181** | -0.118# | 0.040 | -0.111* |
| O. Unstable Relationships (G+K) | 0.060 | 0.137** | 0.026 | 0.088 | -0.003 | -0.070 | 0.100 | 0.054 | 0.109 | -0.041 |

Significantly correlated with the presence of the disclosed risk at the *0.05 or **0.01 level (2-tailed)
Significantly correlated with the absence of the disclosed risk at the #0.05 or ##0.01 level (2-tailed)

**Fig 1. Phi correlations of screening criteria by ACE risk disclosed to MECSH nurse.**

7.3%), and nearly seven times higher for mental health conditions (69.0% cf 9.1%), with 20.7% reporting having three or more ACEs compared with 10.3%. The rates disclosed over time in the right@home program were comparable with the risk profile at enrolment of home visiting clients reported in the US Maternal, Infant, and Early Childhood Home Visiting Program Evaluation (MIHOPE) [32] for substance misuse and domestic violence, but reported rates for depression were much higher than the MIHOPE rate of 30% at enrolment. The right@home rates were considerably higher than those disclosed in screening at three months post enrolment in the Healthy Families America home visiting program HELP study (for example, the HELP study reported a screening prevalence of 13% for maternal depression and 11% for intimate partner [domestic] violence) [10].

The variance in the population rates and those disclosed in other studies can likely be attributed to differences in the timing of and tools used for screening, and this current study did not seek information on whether the disclosed risk was validated through clinical measurement. The retrospective clinical audit used to identify disclosed risks only captured those that were documented in client notes, and may not therefore include all, or minor issues that were not considered as needing to be recorded, and may still underestimate the prevalence of risk. Uniquely this study captured disclosure of risk at any time over the more than two year right@home program intervention with each family. The high risk prevalence may thus be a more accurate picture of the ACE environment for young children in adversity than screening, which provides point prevalence only, or retrospective parental/adult recall such as used in the original ACEs study [1] or the more recent study of parents of children aged 0–17 [2]. Home visiting over a sustained period of time enables practitioners to establish and develop effective trusting therapeutic relationships with parents [33], which may support disclosure.

Importantly, the non-sensitive screening questionnaire identified this high risk population, and also identified a maternal population where the risk was sited with the partner or family, rather than the mother. For example, disclosure of substance misuse was more likely associated with partner or family than with the mother. Understanding of the prevalence of disclosed partner and family risk is needed as these contribute to the overall setting for children's experiences and can contribute to those experiences being adverse. Typically, however, sensitive risk

factor screening in pregnancy is predominantly focussed on risk associated with the mother rather than the broader environment for the child/ren.

Smoking in pregnancy was correlated with substance misuse, both in the mother and also their partner/family. There is increasing evidence from animal, and now human studies that smoking, particularly when commenced in adolescence for females, is a precursor to subsequent alcohol and other drug misuse, and also that concurrent smoking amplifies the negative health impacts of alcohol and other drug misuse (see for example Cross et al [34]).

Not living with another adult (rather than a reported lack of support) was associated with domestic violence and unstable relationships, and major maternal mental health conditions. Recent studies have highlighted the relationship between women's mental health, experience of domestic/intimate partner violence and unstable housing, homelessness and isolation [35, 36]. Maternal anxious mood in the two weeks prior to screening was, as expected, correlated with anxiety and depression. The screening questions used the Matthey General Mood Questionnaire, which has been demonstrated to have better concordance with DSM anxiety disorder than other (more lengthy and intrusive) measures such as the Edinburgh Depression Scale, or Hospital Anxiety and Depression Scale [30, 37]. The study population could have thus been expected to include a high prevalence of mothers with anxiety.

Interestingly, being a young mother (aged <23), a recruitment criterion for other commonly implemented home visiting programs (see for example Roblings et al Building blocks [38]), was very weakly associated with only maternal experience of domestic violence, and thus may not, in this population, be a sufficient indicator of the need for additional support. Poverty (notably more-so than never having had a job) as determined by receipt of state benefits, which is another common home visiting eligibility criterion (see Home Visiting Evidence of Effectiveness–HomVEE [39]), was associated with domestic violence and involvement with child protection services, and substance misuse in the partner or family (but not the mother). The association between non-completion of secondary education and less disclosure of maternal anxiety, depression and mental health conditions was unexpected. Further research is needed to understand this association, particularly as poorer maternal education is reported in many studies to be associated with poor child health and development outcomes [40].

## Limitations

These results cannot be generalized to population prevalence as the data were limited to those women who disclosed two or more screened-for sociodemographic factors in the context of recruitment for a sustained nurse home visiting trial. Also, disclosure information was obtained from retrospective audit by the nurses, and reliant on the quality of their documentation and/or recall and may not have been comprehensive. The nurses did not note whether the disclosures were directly reported by the mother and/or observed by the nurse, and no independent verification of the noted risks (for example, mental health diagnosis confirmation) were sought by the researchers. However, a large number of serious risks were identified consistent with the expected vulnerabilities within a population who could benefit from sustained and intensive support. It is also possible that the intervention may have prevented or mitigated risks, such that the prevalence reported here, albeit high, may underestimate that of a non-intervention population at risk. The correlations between individual screening criteria and disclosed risks were small.

## Conclusion

Children's exposure to ACEs has been recognized by WHO as a significant public health issue. This study showed a high prevalence of ACE risks in the screened population who received the

right@home visiting intervention and demonstrated small, albeit significant patterns of correlations between individual sociodemographic screening factors and disclosed risks. Sociodemographic screening in pregnancy was undertaken using non-sensitive questions that women were comfortable to answer in a public waiting room and identified this high risk population. Certain screened risks were correlated with a number of ACE risk disclosures in this population, with maternal smoking in pregnancy and not living with another adult correlated with multiple ACE risks. Young maternal age was not correlated with the disclosed risks, however, receipt of state benefits (poverty) was correlated with elements of complex family environments; domestic violence, family substance misuse and involvement with child protection services. Although the correlations were significant but weak, screening for these characteristics may reduce dependence on the unreliability of maternal risk disclosure in first or minimal contact screening, and enhance identification of families to engage in interventions to prevent negative consequences of children's exposure to adverse experiences.

## Supporting information

**S1 Checklist. STROBE statement—Checklist of items that should be included in reports of cohort studies.**
(DOC)

## Acknowledgments

The "right@home" sustained nurse home visiting trial is a research collaboration between the Australian Research Alliance for Children and Youth (ARACY); the Translational Research and Social Innovation (TReSI) Group at Western Sydney University; and the Centre for Community Child Health (CCCH), which is a department of The Royal Children's Hospital and a research group of Murdoch Children's Research Institute. We thank all families, the research assistants, and nurses and social care practitioners who worked on the right@home trial, the antenatal clinic staff at participating hospitals who helped facilitate the research, and the Expert Reference Group for their guidance in designing the trial. The MECSH® program is a registered trademark of UNSW Australia and Western Sydney University is the authorised licence holder.

## Author Contributions

**Conceptualization:** Lynn Kemp, Fiona Byrne, Sharon R. Goldfeld.

**Data curation:** Sheryl A. Scharkie.

**Formal analysis:** Lynn Kemp, Emma L. Elcombe.

**Methodology:** Lynn Kemp.

**Project administration:** Tracey Bruce.

**Writing – original draft:** Lynn Kemp.

**Writing – review & editing:** Tracey Bruce, Emma L. Elcombe, Fiona Byrne, Sheryl A. Scharkie, Susan M. Perlen, Sharon R. Goldfeld.

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
