## [Decision Letter · Decision Letter 0]

9 Sep 2021

PONE-D-21-06995Identification of families in need of support: correlates of adverse childhood experiences in the right@home sustained nurse home visiting programPLOS ONE

Dear Dr. Kemp,

Thank you for submitting your manuscript to PLOS ONE. After careful consideration, we feel that it has merit but does not fully meet PLOS ONE’s publication criteria as it currently stands. Therefore, we invite you to submit a revised version of the manuscript that addresses the points raised during the review process.

Please note that as per our publication criteria, PLOS ONE requires that all experiments, statistics and other analyses are performed to a high technical standard, described in sufficient detail and adhere to appropriate reporting guidelines and community standards. Conclusions must be presented in an appropriate fashion and be supported by the data (Please see http://journals.plos.org/plosone/s/criteria-for-publication).

Your manuscript has been assessed by two reviewers who have raised overlapping concerns with the statistical analysis performed and ability of the results to support your conclusions. The reviewers have clearly outlined these concerns so I will not repeat them here. However, I do encourage you to carefully revise your conclusions of the manuscript in light of the concerns raised; statistical significance should not be equated with effect size - as you note in your results section all significant correlations reported in your manuscript were weak. It is therefore not appropriate to indicate that 'Smoking in pregnancy was particularly correlated with substance misuse' (line 264) or indicate that variables shown to have statistical significant association act as a proxy for one another (conclusions). Please bear this in mind as you carefully revise your manuscript to respond to all of the reviewer's concerns. 

We look forward to receiving your revised manuscript.

Kind regards,

George Vousden

Division Editor

PLOS ONE

Journal Requirements:

3. Thank you for stating the following in the Financial Disclosure section: "This work is supported by the Victorian Department of Education and Training, the Tasmanian Department of Health and Human Services, the Ian Potter Foundation, Sabemo Trust, Sidney Myer Fund, the Vincent Fairfax Family Foundation, and the National Health and Medical Research Council (NHMRC, 1079418). The funders had no role in study design, data collection and analysis, decision to publish, or preparation of the manuscript."

We note that you received funding from a commercial source: Ian Potter Foundation and Sabemo Trust

4. Your abstract cannot contain citations. Please only include citations in the body text of the manuscript, and ensure that they remain in ascending numerical order on first mention.

Reviewers' comments:

Reviewer's Responses to Questions

**Comments to the Author**

1. Is the manuscript technically sound, and do the data support the conclusions?

Reviewer #1: Partly

Reviewer #2: No

2. Has the statistical analysis been performed appropriately and rigorously? 

Reviewer #1: I Don't Know

Reviewer #2: I Don't Know

3. Have the authors made all data underlying the findings in their manuscript fully available?

Reviewer #1: Yes

Reviewer #2: Yes

4. Is the manuscript presented in an intelligible fashion and written in standard English?

Reviewer #1: Yes

Reviewer #2: Yes

5. Review Comments to the Author

Reviewer #1: Important note: This review pertains only to ‘statistical aspects’ of the study and so ‘clinical aspects’ [like medical importance, relevance of the study, ‘clinical significance and implication(s)’ of the whole study, etc.] are to be evaluated [should be assessed] separately/independently. Further please note that any ‘statistical review’ is generally done under the assumption that (such) study specific methodological [as well as execution] issues are perfectly taken care of by the investigator(s). This review is not an exception to that and so does not cover clinical aspects {however, seldom comments are made only if those issues are intimately / scientifically related & intermingle with ‘statistical aspects’ of the study}. Agreed that ‘statistical methods’ are used as just tools here, however, they are vital part of methodology [and so should be given due importance].

COMMENTS: Though the use of non-parametric Spearman’s ‘Correlation coefficient (ρ)’ {instead of parametric ‘Pearson’s} is appreciated, as most of measures/tools used are [though appropriate] yield data that are in ‘ordinal’ level of measurement and then application of suitable non-parametric test(s) is/are indicated/advisable even if distribution may be ‘Gaussian’ (i.e., normal)]. Moreover, most of the ‘Spearman correlation Coefficients’ reported in Table 3 [Spearman correlations of screening criteria by disclosed to MECSH nurse] are numerically very small {though sometimes significant}. In this context, please read the following:

Statistical test usually used to assess significance of Pearson’s ‘Correlation coefficient (r)’ is ‘t’ [where t = { r � [(n-2) / (1-r2)] }for df=n-2, n is sample size] and here Ho is that the population/standard value of ‘r’ is zero. You need r=0.878 to be significant at 5% when n=5 but you need r=0.273 only, if n=50 & you need r=0.088 only, if n=500. Because ‘P-value’ heavily depends on sample size, it is customary to use the (available in most text books on ‘Biostatistics’ or on ‘www/net’) guidelines very strongly suggesting] to consider an absolute value of ‘Correlation coefficient for interpreting positive or negative correlations (and do not rely only on corresponding ‘P’-value but also consider an absolute value of ‘Correlation coefficient’). [This argument is equally applicable to non-parametric Spearman’s ‘Correlation coefficient (ρ)’ as well.]

This is pasted from one standard textbook on ‘Research Methodology’ and I am sure that the authors already know these things, however, it is very essential to keep the limitations in mind while interpreting results. In fact, I wonder “how even ‘Spearman’s Correlation coefficients (ρ)s’ are calculated as most of the variables are (seems to be as per lines 167-8: Spearman correlations were conducted to test the association between the presence/absence of screening factors and presence/absence of disclosed ACE risks.) ‘binary/dichotomous’ [where only biserial or point biserial correlation coefficients are appropriate”. Will you please explain? (may be correctly done, but explanation is essential).

Major part (lines 309-317: In this population, maternal smoking in pregnancy could be considered to be acting as a proxy for substance misuse, involvement with child protection services, mental health issues and family violence, suggesting that in similar populations particular consideration be given to additional and sustained support for mothers who are smoking in pregnancy. Not living with another adult was a proxy for unstable and violent relationships and major mental health issues; and anxious mood in the weeks prior to screening in pregnancy as a proxy for anxiety/depression and mental health issues during the subsequent two years. Maternal age was not a useful indicator of risk, however, receipt of state benefits was a proxy for a complex family environment) of ‘conclusion’ section is not really a [rather may/cannot be the] part of conclusion. Opinion(s) [may be even if learned from this experience], in my opinion, are not to be included in ‘conclusions’ of the study.

Overall, the presentation is confusing with respect few points {example – lines 139-140: 363 women were allocated to the trial intervention arm, and 352 commenced the intervention program what is the relevance of such grouping? No group comparison is clearly seen anywhere later. How the ‘allocation’ was performed is not given. Moreover, at few places {example - lines 167-8: Spearman correlations were conducted to test the association between the presence/absence of screening factors and presence/absence of disclosed ACE risks as we never conduct the correlations} construction of a sentence needs to be checked, I guess.

Limitations of the study are not discussed in a separate section (expected). Lines 295-97 mention only one limitation [These results cannot be generalized to population prevalence as the data were limited to those women who disclosed two or more screened-for sociodemographic factors in the context of recruitment for a sustained nurse home visiting trial.] though, which is not enough. {Does that mean {according to authors} there are none?} In my opinion, though it is a straight-forward simple study (not many methodological issues), contributes a marginal/minimal.

As pointed out in ‘important note’ above “This review pertains only to ‘statistical aspects’ of the study and so ‘clinical aspects’ should be assessed separately/independently. In my opinion, to rescue this article (which is difficult but not impossible), lot of re-vision is needed. Therefore, just short of ‘rejection’, I am recommending “Major Revision”.

Reviewer #2: PLOS One

PONE-D-21-06995

The purpose of this manuscript was to examine the association between sociodemographic measures collected at entry to a home visiting program and risks were identified based on disclosures to the nurse and measured in subsequent nurse case files. Screening was reported by the mother at the antenatal clinic and risks were identified by nurses through a retrospective case audit following HV services up to age 2 years of the child. This unique dataset explores an important topic in the field. However, I came away with totally different interpretation of the data presented. Specifically, the very weak associations (although statistically significant) between screening measures and subsequent risks imply that these are not good measures from a sensitivity/specificity perspective in identifying families who may benefit from sustained home visiting.

The introduction overall is well-written and clear. The authors capture the main issues the field is grappling with. The authors make a compelling case that a non-sensitive screening tool with high specificity would be ideal for many reasons. I also suggest the authors cite work in the US that utilized data from birth certificates that are highly predictive of later child maltreatment. However, I am not sure that the items in the screening tool presented here are actually non-sensitive since they include socially stigmatized domains such as poverty, maternal smoking, and mental health concerns that at least in most “Western” contexts would be subject social desirability.

There is another limitation to the approach that is not considered, it appears that the ACEs are risks disclosed to the nurse, while the screening tool is self-reported by the mother. It is not clear whether these disclosures had to be reported directly from the mother to the nurse, or if the nurse could observe a risk and this counts as a disclosure. For example, if a nurse knew that a family was involved with CPS but the mother never disclosed this detail, how would the nurse document this in the spreadsheet? We then wade into territory regarding subjective/objective assessments and agreement between different reporters. But this is a fairly minor issue.

The major concern regards the “weak” correlations, as stated by the authors. In the conclusions the authors provide an interpretation that since the association between smoking and substance misuse was .273, that this is “particularly correlated”. I would disagree. Similar the correlated between anxious mood and anxiety was .186, this is quite a weak association. It is then suggested that these are now “proxies” for actual risks which to me is a very dangerous conclusion. Although the stakes in false-positives or negatives for this type of prediction are low (worst case you get offered a home visiting program), we should not equate a low association with a valid predictor.

I was also curious why multivariable models were not used to see how screening measures predicted risks, while controlling for all of these factors, but perhaps this was not the goal of the research.

6. PLOS authors have the option to publish the peer review history of their article (what does this mean?). If published, this will include your full peer review and any attached files.

Reviewer #1: No

Reviewer #2: No

---

## [Author Response · Author response to Decision Letter 0]

23 Jan 2022

We would like to thank the editor and reviewers for their helpful and constructive review of our paper. We have documented our responses in the covering letter file, which includes a table of our responses for each reviewer's comments. Thank you.

---

## [Decision Letter · Decision Letter 1]

8 Jul 2022

PONE-D-21-06995R1Identification of families in need of support: correlates of adverse childhood experiences in the right@home sustained nurse home visiting programPLOS ONE

Dear Dr. Kemp,

Thank you for submitting your manuscript to PLOS ONE. After careful consideration, we feel that it has merit but does not fully meet PLOS ONE’s publication criteria as it currently stands. Therefore, we invite you to submit a revised version of the manuscript that addresses the points raised during the review process.

Specifically, we require that conclusions are presented in an appropriate fashion and are supported by the data. We still have concerns about some of the statements in your conclusion. For instance, you indicate that "Smoking in pregnancy was particularly correlated with substance misuse" (line 256). We would suggest removing "particularly" in this case as only weak correlation is reported and we feel your data does not support the conclusion here. In addition, you indicate that "Sociodemographic screening in pregnancy using non-sensitive questions in this study appropriately identified a population of families with risks for ACEs, for whom sustained home visiting support may be of benefit to mediate the impact of risk and improve outcomes." (line 313-315). We do not feel this is supported by your data -  only (weakly) correlation was presented for the sociodemographic characteristics with ACEs, and that does not "identify" population of families with risks for ACEs. Please ensure that all the statements in Conclusion are supported by your data presented and not overstate your conclusions.Finally, please provide your responses to the first reviewer's comments below.

We look forward to receiving your revised manuscript.

Kind regards,

Jianhong Zhou

Staff Editor

PLOS ONE

Journal Requirements:

"The MECSH® program is a registered trademark of UNSW Australia and from 2016 for the duration of 5 years is being sublicensed to Western Sydney University."

Reviewers' comments:

Reviewer's Responses to Questions

**Comments to the Author**

1. If the authors have adequately addressed your comments raised in a previous round of review and you feel that this manuscript is now acceptable for publication, you may indicate that here to bypass the “Comments to the Author” section, enter your conflict of interest statement in the “Confidential to Editor” section, and submit your "Accept" recommendation.

Reviewer #1: All comments have been addressed

2. Is the manuscript technically sound, and do the data support the conclusions?

Reviewer #1: (No Response)

3. Has the statistical analysis been performed appropriately and rigorously? 

Reviewer #1: (No Response)

4. Have the authors made all data underlying the findings in their manuscript fully available?

Reviewer #1: (No Response)

5. Is the manuscript presented in an intelligible fashion and written in standard English?

Reviewer #1: (No Response)

6. Review Comments to the Author

Reviewer #1: COMMENTS: Since all of the comments made on earlier draft by me (and hopefully by other respected reviewers also) were/are attended positively, I recommend the acceptance because the manuscript now has achieved acceptable level, in my opinion.

Nevertheless, I wish/want to let authors note that, even if [while the assessment of correlation in dichotomous vs dichotomous case] Pearson’s r, Spearman’s Rho, and Phi all produce the same results {though the SPSS output they have kindly provided, are not visible}, always report the most applicable ones. And also note that figure(s) not [always] the substitute of table(s). They are, generally, not ‘alternatives’ but ‘complementary’.

7. PLOS authors have the option to publish the peer review history of their article (what does this mean?). If published, this will include your full peer review and any attached files.

Reviewer #1: **Yes: **Dr. Sanjeev Sarmukaddam

---

## [Author Response · Author response to Decision Letter 1]

11 Aug 2022

The response to reviewer and editor comments are provided in the file "Response to Reviewers".

---

## [Editor Report · Decision Letter 2]

19 Sep 2022

Identification of families in need of support: correlates of adverse childhood experiences in the right@home sustained nurse home visiting program

PONE-D-21-06995R2

Dear Dr. Kemp,

We’re pleased to inform you that your manuscript has been judged scientifically suitable for publication and will be formally accepted for publication once it meets all outstanding technical requirements.

Kind regards,

Jianhong Zhou

Staff Editor

PLOS ONE
---

## [Editor Report · Acceptance letter]

22 Sep 2022

PONE-D-21-06995R2 

Identification of families in need of support: correlates of adverse childhood experiences in the right@home sustained nurse home visiting program 

Dear Dr. Kemp:

I'm pleased to inform you that your manuscript has been deemed suitable for publication in PLOS ONE. Congratulations! Your manuscript is now with our production department. 

Kind regards, 

on behalf of

Jianhong Zhou 

Staff Editor

PLOS ONE